# Identifying predictors and assessing causal effect on hypertension risk among adults using Double Machine Learning models: Insights from Bangladesh Demographic and Health Survey

**Probir Kumar Ghosh** [ID]*, **Md. Aminul Islam, Md. Ahshanul Haque, Md. Tariqujjaman, Novel Chandra Das, Mohammad Ali, Md. Rasel Uddin, Md. Golam Dostogir Harun**

International Centre for Diarrhoeal Disease Research, Dhaka, Bangladesh

* probir@icddrb.org

## Abstract

### Background

Hypertension poses a significant public health challenge in low- and middle-income countries. In Bangladesh, the Health Population and Nutrition Sector Development Program has shown effectiveness in resource-limited settings. Estimating causal relationships on hypertension while adjusting for nonlinear observed confounders in adult population is complex. This study aims to identify predictors of hypertension, and explore observational causal inference on hypertension.

### Methods

The hypertension data was analyzed using Bangladesh Demographic and Health surveys data from the 2011 and 2022. We used 11,815 individuals aged 34 years and above. Hypertension was defined as a systolic blood pressure of > 140 mm Hg and/or a diastolic blood pressure of > 90 mm Hg and/or having a history of hypertension. We used logistic regression, Random forest model, Double Machine Learning (DML), and Shapley Additive exPlanations (SHAP) based on a pre-defined causal structure.

### Results

The dataset included 11,815 individuals, and the prevalence of hypertension was 38.40%. The average age of individuals was 52.76 years (SD: 12.97), and 6826 (58.77%) were male. The Random forest model achieved 93% accuracy, with evaluation f1-scores of 95% for non-hypertension and 91% for hypertension, and identified older age, female gender, urban residency, workers, wealthier, self-awareness, and excessive body weight as key predictors of hypertension. The individual conditional expectation and SHAP plots reveal that age, and body mass index (BMI) are nonlinear relation with hypertension. The crude OR between excessive body weight and

**Data availability statement:** All relevant data used in this study are publicly available from the Demographic and Health Surveys (DHS) Program website: https://dhsprogram.com/data/available-datasets.cfm.

**Funding:** The author(s) received no specific funding for this work.

**Competing interests:** The authors have declared that no competing interests exist.

hypertension was 2.24 (95%CI: 2.07 – 2.42). Adjusted for age, sex, socioeconomic status (SES), and self-awareness, the OR was 1.97 (95%CI: 1.79 – 2.17), and using de-biased method, it was 1.30 (95%CI: 1.17 – 1.43).

## Conclusion

The study highlights important predictors of hypertension, including age, sex, residency, and socioeconomic status (SES), self-awareness and body weight. The machine learning model achieved an accuracy of 93% in predicting hypertension. The de-biased methods provided a more refined risk estimate. Age and excessive body weight were found to significantly contributed to hypertension, demonstrating complex interactions and varying marginal effects across different levels of these factors. Awareness programs and targeted interventions are vital to effectively reduce excessive body weight and prevent hypertension.

## Author summary

Hypertension is an increasing public health challenge in low- and middle-income countries, including Bangladesh, where health interventions often face resource constraints. This study utilized data from the 2011 to the 2022 Bangladesh Demographic and Health Surveys (BDHS) to identify key predictors and examine the causal relationship between excessive body weight and hypertension among adults aged 34 years and older. We applied logistic regression, Random Forest, Double Machine Learning (DML), and SHAP (Shapley Additive Explanations) methods within a predefined causal framework. The Random Forest model achieved a predictive accuracy of 93%, identifying older age, excessive body weight, urban residence, higher wealth, employment status, and self-awareness as significant predictors of hypertension. To obtain unbiased estimates of causality, we used DML, which yielded a de-biased causal effect, providing the causal relationship between excessive body weight and hypertension. The adjusted odds ratio for excessive body weight on hypertension decreased after controlling for key confounders and estimation bias. Additionally, Individual Conditional Expectation (ICE) analysis revealed nonlinear and interaction effects involving age, sex, and self-awareness in modifying the influence of body weight on hypertension risk. These findings underscore the need for targeted awareness programs and policy-driven interventions focused on adult weight management and lifestyle modification.

## Introduction

Hypertension is a widespread health concern, with approximately 1.28 billion people affected globally, majority in low-middle income countries (LMICs) [1]. As a leading non-communicable disease (NCD), hypertension contributes to global health

challenges, especially affecting among older populations, increasing the risk of heart diseases and rate of premature death and disabilities [2,3]. The impact of hypertension is particularly pronounced in the low-middle income countries due to their nutrition status [4], socioeconomic characteristics [5], and epidemiological transitions [6,7]. Moreover, studies have shown complex interrelationships between hypertension, excessive body weight and diabetes in these regions [8–11]. Additionally, various predictors can predict the likelihood of developing hypertension in future, serving as an early warning system for NCD [8,12,13].

Previous studies have highlighted the significant role of other NCDs, including age, sex, nutritional status, diabetes and other covariates in predicting hypertension in the future [14–16]. A few studies have assessed the causal effect of predictors on adult hypertension risk. The Bangladesh Demographic, and Health Surveys in 2011, national population-based surveys, have shown the prevalence of NCDs, awareness and hypertension prevention. However, the methods used in these surveys was not robust, leading to potential biases of predictors in the results. The findings indicated that age, sex residency were key predictors of adult hypertension among adults over 34 years, alone with data on hypertension measurement, self-awareness, and control [17]. Prior study relied on conventional regression models to identify potential risk factors for hypertension. While these models are widely used, they have important limitations in causal inference— particularly their inability to adequately control for confounding bias when estimating the effects of predictors [18]. Unlike modern causal inference techniques, such as Double Machine Learning (DML) used in our study, conventional regression approaches do not incorporate cross-fitting or data-driven selection of nuisance parameters, which are essential for reducing estimation bias and improving robustness [19].

It is well-established that a causal pathways of predictors influence the hypertension risk, with socioeconomic status (SES) factor, age, sex, self-awareness, body weight, and diabetes being key predictors of hypertension [18,20,21]. After adjusting for confounders, studies have shown that excessive body weight and self-aware are significantly associated with hypertension.[18,20,22–24]. In addition, nutritional status remains a major predictor of hypertension and is closely linked to behavioral and economic predictors [25]. The dissimilarities between rich and poor are evident in their social habits, food habit, lifestyle, and access to healthcare [26]. In wealthier communities, individuals often experience a higher burden of non-communicable diseases due to sedentary and unhealthy lifestyle [27,28].

Many previous studies have attempted to estimate causal effects, exploring how predictors of interest causally affects the outcome [14,18,28,29]. For example, researchers often seek to determine if excessive body weight causes hypertension [14,18]. To understand a causal relationship between excessive body weight and developing hypertension, observational data is typically used, as experimental interventions are often infeasible, unethical, and costly [30]. However, the assumption that all confounders are observed and properly adjusted for is one of the biggest challenges in casual inference applications [31–33]. Supervised boosting machine learning models can relax some of these assumptions, particularly in nonlinear forms, and handle high dimensional data. While machine learning model is designed for prediction, it primary goal differs from causal inference, as it does not directly interpret causal effects [13,34,35]. A Double Machine Learning (DML) is a leading method that utilizes machine learning for observational causal inference without requiring parametric model assumptions [30,32]. This approach adjust for confounders to provide unbiased estimates, even with numerous confounders and complex nonlinear relationships [19,36].

Quantitative risk assessments play a crucial role in understanding the predictors of hypertension and their impact on health outcomes [37–39]. These assessments have identified preventive measures to mitigate adverse health effects. Some key predictors directly influence hypertension while others act indirectly by confounding causal links [18,38]. For instance, Health Population and Nutrition Sector Development Program (HPNSDP) in Bangladesh prioritized increasing self-awareness of non-communicable diseases (NCDs).This focus was driven by achievements in increasing life expectancy, reducing malnutrition and health outcomes [40]. Despite its success in improving adult health, addressing the growing burden of hypertension remains a significant challenge [18,38]. This analysis aims to investigate predictors of hypertension and explore causal relationships between predictors and hypertension among older adults in Bangladesh.

## Methods

### Ethics statement

No ethics approval was required for this study because it used secondary analysis of publicly available Bangladesh Demographic and Health Surveys (BDHS) data. The protocols for adult hypertension were reviewed and approved by both the Inner City Fund (ICF) Institutional Review Board ethics committee and the Bangladesh Medical Research Council (BMRC). The BDHS team also obtained informed written consent from all participants before the interview.

Since 2011, the Health, Population, and Nutrition Sector Development Program (HPNSDP) in Bangladesh has prioritized increasing life expectancy at birth, reducing non-communicable diseases (NCDs), and combating malnutrition [41,42]. As part of this initiative, the BDHS 2011 included baseline indicators for adult NCDs, targeting individuals aged 34 years and older. Preliminary findings from BDHS 2011 revealed that one in three adults in this age group were hypertensive [17].

To address the growing focus on NCDs, the 4th Health, Population, and Nutrition Sector Program (HPNSP) during 2016–2021 introduced a dedicated component to improve life expectancy, expand immunization coverage, reduce malnutrition, and tackle the dual burden of communicable and non-communicable diseases associated with Bangladesh's epidemiological transition [43]. Subsequent surveys—BDHS 2017–18 and BDHS 2022—collected updated NCD data to evaluate adult health outcomes [17].

While significant progress has been made in improving children's health and nutrition outcomes, adult health remains a challenge. Each BDHS survey collected NCD data from adults aged 34 years and older in one-third of the surveyed households, comprising 18,000 households in 2011, 20,160 households in 2017–18, and 30,330 households in 2022. These surveys employed a two-stage stratified cluster sampling method. In the first stage, 675 clusters were selected from rural and urban areas across eight divisions, followed by systematic selection of 30 households per cluster in the second stage. The methodologies and sampling strategies for these surveys are detailed elsewhere [17,44].

NCD-related biomarkers and other relevant data were collected during the surveys. Blood pressure (BP) measurements were taken from all adults over 34 years using digital oscillometric devices with automatic upper-arm inflation and pressure release, conducted by trained health workers during interviews. Systolic and diastolic BP values were recorded in millimeters of mercury (mmHg). Hypertension was defined as systolic $BP \geq 140$ mmHg and/or diastolic $BP \geq 90$ mmHg, in accordance with the National Guidelines for Management of Hypertension in Bangladesh [45].

In addition to NCD data, individual and household characteristics were analyzed to assess potential confounding variables. These included socioeconomic status (SES), household assets, education, and nutritional indicators such as height and weight. Adult nutritional status was evaluated using body mass index (BMI), calculated as weight in kilograms divided by height in meters squared ($kg/m^2$). BMI categories included underweight ($<18.5\,kg/m^2$), normal weight ($18.5–24\,kg/m^2$), overweight ($25–29.9\,kg/m^2$), and obese ($\geq 30\,kg/m^2$).

Causal inference is a statistical and mathematical approach used to determine the true effect of one variable on another. It helps assess whether an exposure variable has a causal impact on a specific outcome variable. In this study, we employed Double Machine Learning (DML)—a flexible and widely used method that integrates causal inference techniques with machine learning algorithms. The nuisance parameters—namely, the outcome and the exposures model—were both estimated using Random Forest algorithms. These models were trained to predict the outcome and exposure assignment, respectively, based on covariates. We implemented K-fold cross-fitting (with $K = 10$) to avoid overfitting and ensure valid inference, as recommended in the original DML framework. In this procedure, the dataset was partitioned into K folds; the nuisance functions were estimated on K-1 folds and used to compute the orthogonalized score on the held-out fold. This process was repeated across all folds, and the estimates were aggregated to produce the final causal effect estimate [19]. A crucial initial step in any causal inference analysis is identifying the underlying causal relationships between variables, a process known as causal discovery. This step enables us to quantify causal effects and inter-variable

relationships [19,32,41]. A prior study employed a Double Machine Learning (DML) approach to evaluate the effectiveness of COVID-19 vaccines within the framework of the test-negative design. The DML model performed well, effectively accounting for confounding variables and yielding robust estimates of vaccine effectiveness [42]. Using the BDHS adult dataset, we applied the Double Machine Learning/ de-biased model to explain how to uncover causal connections between predictors and the outcome of hypertension.

## Statistical analysis

Descriptive analysis was conducted to summarize the data. For continuous variables, the mean and standard deviation (SD) were calculated, while frequency and proportion were determined for categorical variables. Missing values were excluded from the dataset to ensure analytical reliability and accuracy. This approach minimizes potential biases, enhances predictive power, and improves computational efficiency, thereby contributing to robust and reliable outcomes [43,44].

To predict individuals with hypertension, 25 variables were analyzed, including baseline characteristics, regional data, nutritional status, and physical activity-related variables. Continuous variables were standardized using a scaler, while binary variables were normalized with a MinMax scaler to enhance model accuracy and computational performance [45,46].

Key variables were selected for inclusion in a machine learning classification model. We evaluated the performance of logistic regression, Least Absolute Shrinkage (Lasso), Ridge, XGBoost, Random forest classifier models using training and validation errors to identify the best performing model. Among these, the Random Forest classifier model—an ensemble supervised learning algorithm—demonstrated the best performance and was selected for this purpose. Random forest model generates multiple decision trees during training, introducing randomness to reduce correlations among tree predictions. The classification process aggregates the outputs of individual trees, with the final prediction based on the mode of predicted classes [47]. Machine learning was used to identify predictors of hypertension, followed by logistic regression to examine the associations between hypertension and key predictors [48].

To address class imbalance in the target variable (hypertension vs. non-hypertension), the Synthetic Minority Oversampling Technique (SMOTE) was applied. This mitigated overfitting risks and ensured balanced data representation within the Random Forest (RF) model [49]. The dataset was divided randomly into training (80%) and testing (20%) subsets, maintaining consistent proportions of hypertension cases across both subsets.

The performance of the machine learning model was evaluated using metrics such as F1-score, precision, recall, and overall accuracy. Moreover, the feature importance scores presented in the "mean decrease in impurity" (MDI) method, which is used in decision tree–based models such as Random Forest. This method measures how much each feature reduces the impurity (e.g., Gini impurity or entropy) across all the splits in the trees. A higher mean decrease in impurity indicates that the feature contributes more to making accuracy, and therefore is considered more important. Additionally, Shapley Additive exPlanations (SHAP) and Individual Conditional Expectation (ICE) plots were utilized to interpret the model's predictions, offering insights into the contributions of individual features to hypertension prediction and identifying interactions. SHAP, an additive feature attribution method, quantifies the effect of each feature on the model's output, providing interpretable results [50].

Univariate logistic regression was performed to estimate the odds ratio (OR), reflecting the crude association between predictors and hypertension without accounting for confounders. However, this approach may produce biased estimates due to confounding variables associated with both predictors and outcomes. To achieve more accurate estimates, multivariate logistic regression was applied, adjusting for potential confounders [18,29].

Logistic regression assumes a linear relationship between the log-odds of the outcome and predictors. If this assumption is violated, the results may deviate from the true associations. To address this limitation, a Double Machine Learning model was used [19]. This model captures both linear and non-linear relationships as well as complex interactions in

high-dimensional data, leading to more precise estimates of the observational causal effects of predictors on hypertension, guided by the conceptual framework illustrated in Fig 1. The causal structure presented in Fig 1 is adapted from our previously published study [18], In this study, we constructed a Directed Acyclic Graph (DAG) to represent hypothesized relationships between key variables influencing hypertension among older adults. The DAG reflects domain knowledge and prior epidemiological evidence, identifying variables such as age, sex, socioeconomic status (SES), education level, and body mass index (BMI) as potential confounders in the relationship between behavioral or biological exposures and hypertension outcomes. Variables such as dietary patterns, stress, and genetics, though recognized as important, were excluded from the DAG due to lack of availability in the BDHS dataset. The graph was constructed to satisfy the backdoor criterion, ensuring appropriate adjustment for confounding and allowing valid causal inference. This structure underpins the variable selection for the Double Machine Learning (DML) framework used in this study and serves as a visual guide for understanding the assumptions required to estimate the causal effect of excessive body weight on hypertension. The DAG posits that variables such as age, sex, socioeconomic status, and comorbidities act as confounders in the exposure-outcome relationship and are therefore included in the adjustment set. Other variables, believed to affect the outcome only indirectly through these observed covariates, are excluded based on prior domain knowledge. The structure satisfies the backdoor criterion, allowing for valid identification of the causal effect through appropriate adjustment [31,32].

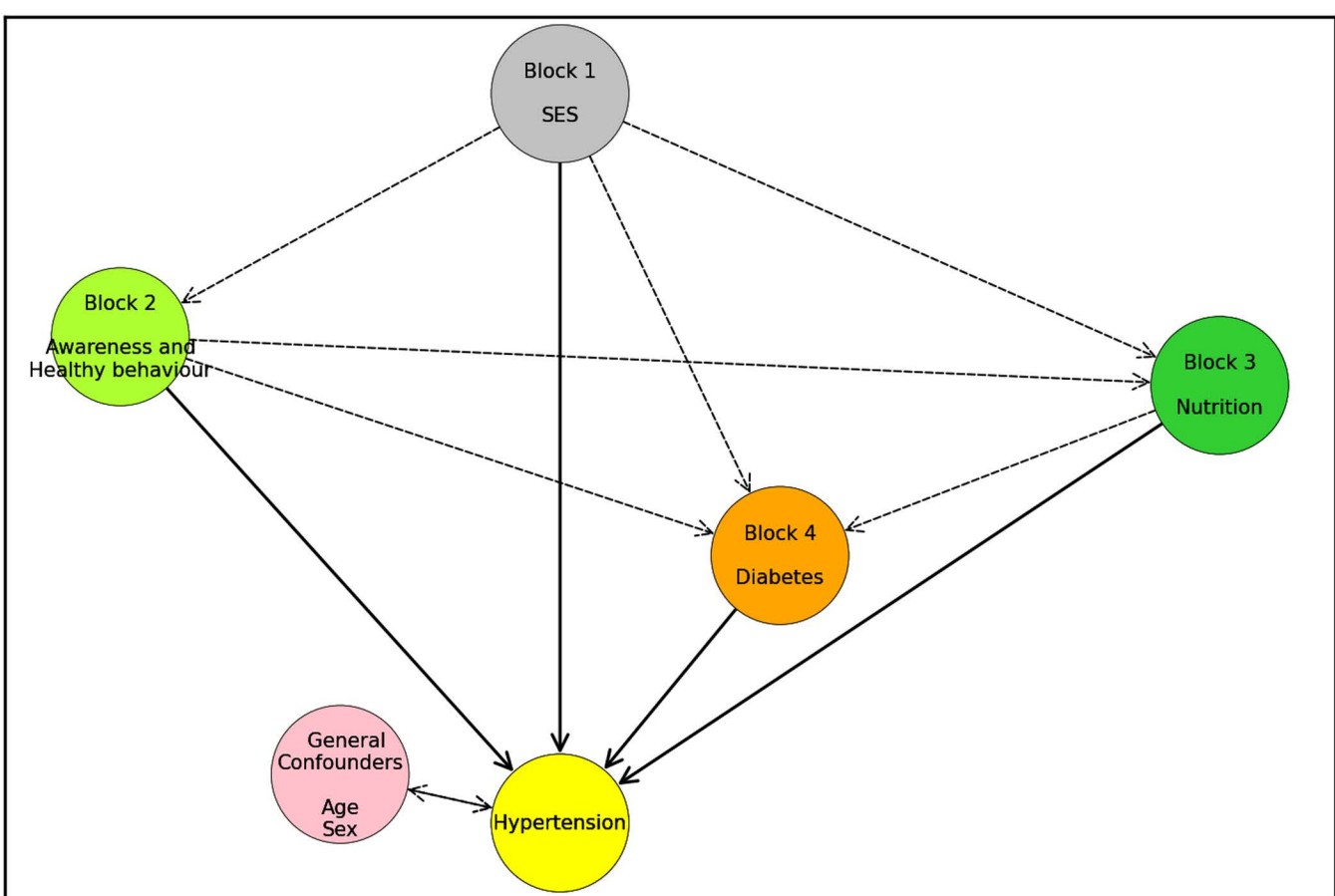

**Fig 1. Causal structure depicts how nutritional status, socio-economic status (SES), awareness, and hypertension prevalence are linked.**
Solid arrows show direct effects. The double arrow represents general confounders as detail elsewhere [18].

Statistical significance was assessed using p-values ≤ 0.05. All data management, visualization, and modeling were performed using Python (version 3.10) and the Scikit-learn library [51].

## Results

### Baseline characteristics

A total of 14,215 adult participants age above 34 years were included in the analytical study. Among them, 26 variables without missing values were available for 11,815 participants and were analyzed them. Table 1 summarizes individual's demographic characteristics, awareness and nutritional data for this analysis. The average (SD) age of participants was 52.76 years (SD: 12.97) and average body mass index (BMI) was 21.66 (SD: 4.08), with more than half being male 6826 (58.77%), and 3918 (33.16%) was urban participants. The no formal education of them was 4918 (41.63%), 3521 (29.80%) of primary, 1798 (15.22%) of secondary, and 1578 (13.36%) of higher educated participants. Regional

**Table 1. Participant's characteristics, socioeconomic status (SES), self-awareness and nutritional status among participants aged 34 year and older in Bangladesh from 2011 to 2022.**

| Characteristics | Mean (SD) | Median (IQR) |
|---|---|---|
| Age of participants in Years | 52.76 (12.97) | 51.00 (18.0 = 42.0-60.0) |
| Body Mass Index (BMI) | 21.66 (4.08) | 21.12 (6.52 = 18.67-24.15) |
| **Characteristics** | **N = 11,815 N** | **%** |
| Male | 6826 | 58.77 |
| Urban | 3918 | 33.16 |
| **Admistrative Division** | | |
| Barishal | 1330 | 11.26 |
| Chattogram | 1618 | 13.69 |
| Dhaka | 1652 | 13.98 |
| Khulna | 1840 | 15.57 |
| Rangpur | 1639 | 13.87 |
| Sylhet | 1545 | 13.08 |
| Mymensingh | 733 | 6.20 |
| Rajshahi | 1458 | 12.34 |
| **Wealth index** | | |
| Poorer | 2295 | 19.41 |
| Poor | 2226 | 18.84 |
| Middle | 2317 | 19.61 |
| Richer | 2316 | 19.60 |
| Richest | 2663 | 22.54 |
| **Educational attainment** | | |
| No formal education | 4918 | 41.63 |
| Primary | 3521 | 29.80 |
| Secondary | 1798 | 15.22 |
| Higher | 1578 | 13.36 |
| **Awareness** | 1837 | 15.55 |
| **worker** | 4293 | 36.34 |
| **Excessive body weight** | 3962 | 33.53 |
| **Diabetes** | 1288 | 10.90 |

distribution of participants showed 1330 (11.26%) in Barishal, 1628 (13.69%) in Chattogram, 1652 (13.98%) in Dhaka, 1840 (15.57%) in Khulna, 1639 (13.87%) in Rangpur, 1458 (12.34%) in Rajshahi, 1545 (13.08%) in Sylhet, and 733 (6.20%) in Mymensingh.

Among the participants, 1,837 (15.55%) were aware of their hypertension status, 4,293 (36.34%) were workers, 1,288 (10.90%) had diabetes, and 3,962 (33.53%) were classified as having excessive body weight. Fig 2 shows that the overall prevalence of hypertension was 38.4% between 2011 and 2022. After applying the Synthetic Minority Oversampling Technique (SMOTE) to balance the training and test datasets and mitigate overfitting, the prevalence of hypertension in both datasets was adjusted to 50%.

## Model performance

Table 2 summarizes the training and validation errors for the Logistic Regression, Lasso, Ridge, XGBoost, and Random forest models. Among these, the random forest classifier model achieved the lowest errors with a training error of 3.0% and validation error of 6.50%. Table 3 and Fig 3 present the precision, recall and confusion matrix, highlighting that the Random Forest model achieved an overall accuracy of 93%. The model's evaluation metrics included F1-scores of 95% for non-hypertension and 91% for hypertension. Precision values were 95% for non-hypertension and 91% for hypertension, while recall scores were 94% and 92% for non-hypertension and hypertension, respectively. On average, the model misclassified 3% of non-hypertension cases and 3% of hypertension cases.

## Importance factors

The Random Forest Classifier model and SHAP values visualization were used to compute the importance of each feature contributing to the classification of participants. These features play a crucial role in building a robust predictive model

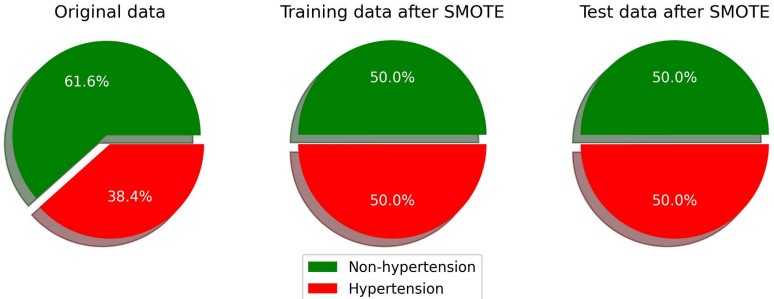

**Fig 2. Hypertension distribution in the original dataset, and balanced class label distribution after applying the Synthetic Minority Oversampling Technique (SMOTE) for training and test data.**

**Table 2. The different models' performance evaluation using original and SMOTE-adjusted data, including training errors, and validation errors based on characteristics, socioeconomic status, self-awareness and nutritional status among participants aged 34 years and older in Bangladesh from 2011 to 2022.**

| Models | Original data | | SMOTE-adjusted | |
|---|---|---|---|---|
| | Training error (%) | Validation error (%) | Training error (%) | Validation error (%) |
| Logistic regression | 29.58 | 29.71 | 26.36 | 30.0 |
| Lasso model | 47.21 | 47.20 | 49.83 | 49.82 |
| Ridge model | 39.59 | 39.63 | 38.60 | 41.43 |
| XGBoost model | 14.62 | 31.57 | 12.59 | 18.47 |
| Random forest model | 3.10 | 9.87 | 3.0 | 6.5 |

**Table 3. Random forest classifier model performance evaluation scores based on characteristics, socioeconomic status (SES), self-awareness and nutritional status among participants aged 34 year and older in Bangladesh from 2011 to 2022.**

| | Classes | Precision (%) | Recall (Sensitivity) (%) | F1-score (%) |
|---|---|---|---|---|
| **Cross-validation (1,000 iteration)** | **Mean** | **91.4** | **92.5** | **91.9** |
| | **Median** | **91.5** | **92.5** | **91.9** |
| | **Standard deviation** | **0.012** | **0.017** | **0.011** |
| Test set | Non-hypertension | 95.0 | 94.0 | 95.0 |
| | Hypertension | 91.0 | 92.0 | 91.0 |
| | **Overall Accuracy** | | | 93.0 |

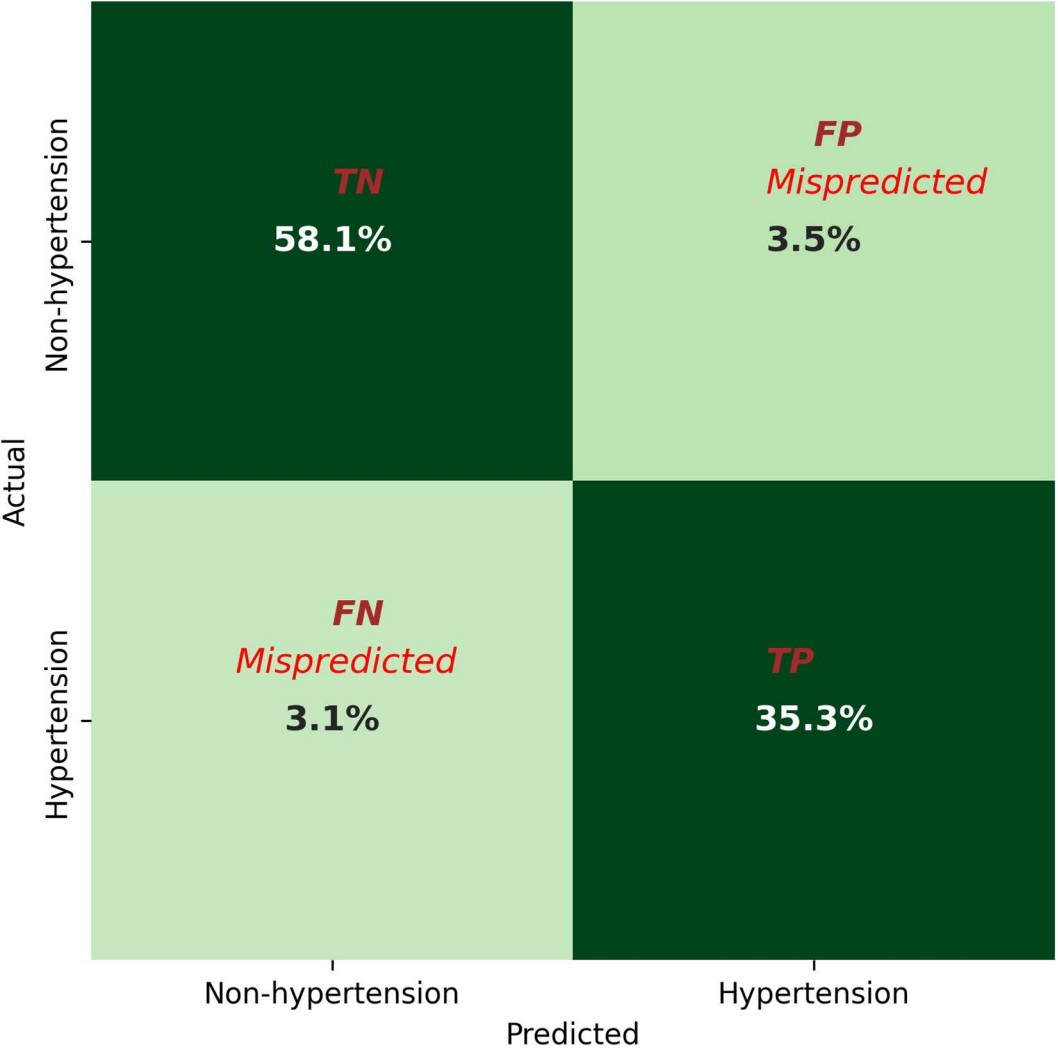

**Fig 3. Confusion matrix (Accuracy = 93%)** is assessing the model's performance, provides True Positives (TP), where the model correctly identifies positive cases as positive; True Negatives (TN), where it accurately predicts negative cases as negative; False Positives (FP), where negative cases are incorrectly classified as positive; and False Negatives (FN), where positive cases are mistakenly classified as negative.

for identifying hypertension. Fig 4 illustrates the global factor importance scores, identifying the most influential factors for hypertension. Body mass index (BMI) (31.28%) and age (22.44%) were the most significant contributors, followed by awareness (5.53%), male gender (3.79%), urban residency (3.12%), worker status (2.35%), and diabetes (2.02%). In contrast, factors such as educational attainment and geographical regions had lower significance, each contributing less than 2%.

## Decision tree

Fig 5 depicts a decision tree that captures participant heterogeneity in hypertension classification. With a depth of four, the tree uses a series of binary decisions to classify individuals as hypertensive or non-hypertensive. For example, individuals over 40 years old with excessive body weight, residing in Chattogram, and self-aware of their hypertension status are likely classified as hypertensive. On the other hand, a male participant with no hypertension awareness and a body mass

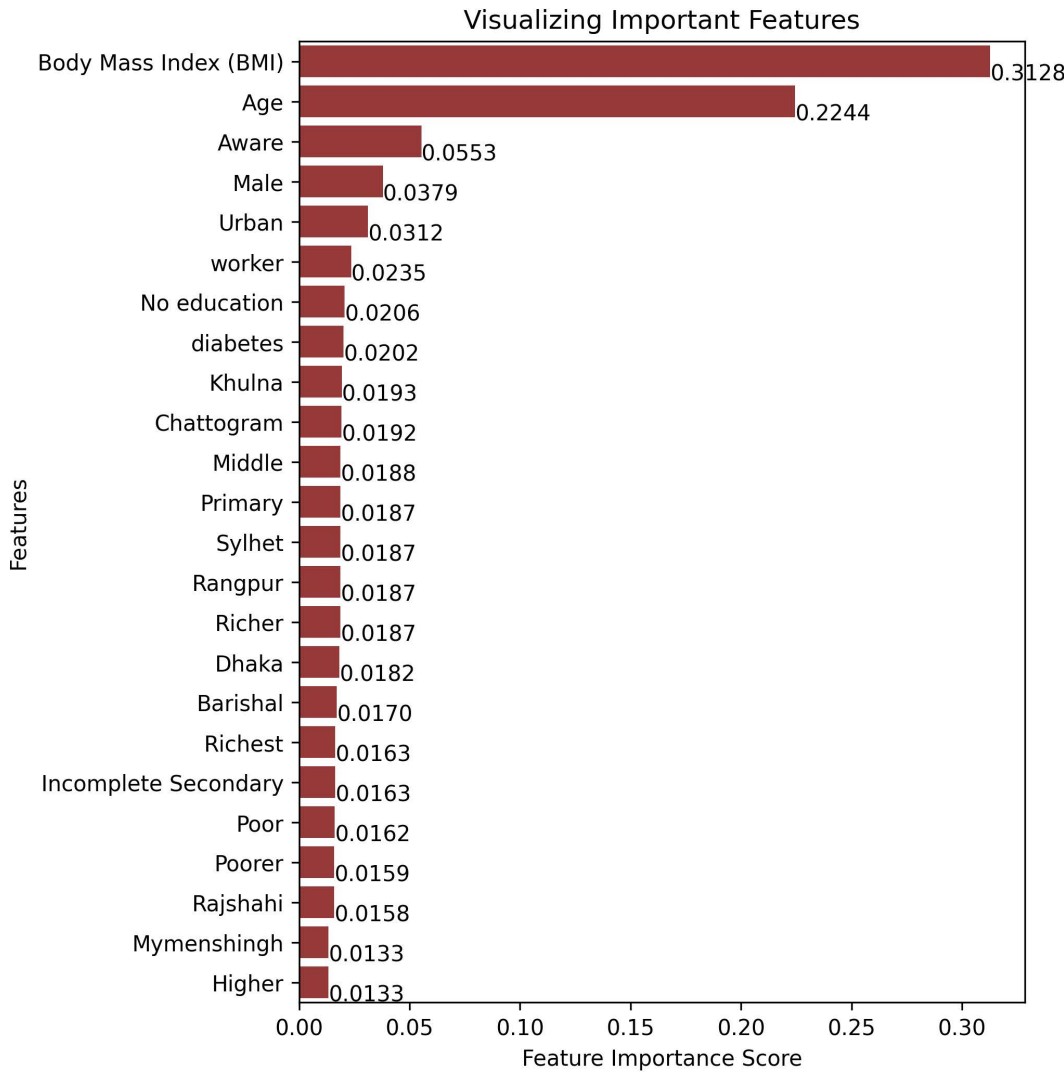

**Fig 4. Features importance reflect how much a feature contributes to model prediction.** Some features with large scores are the most important features, while some features with low scores are less important features for prediction.

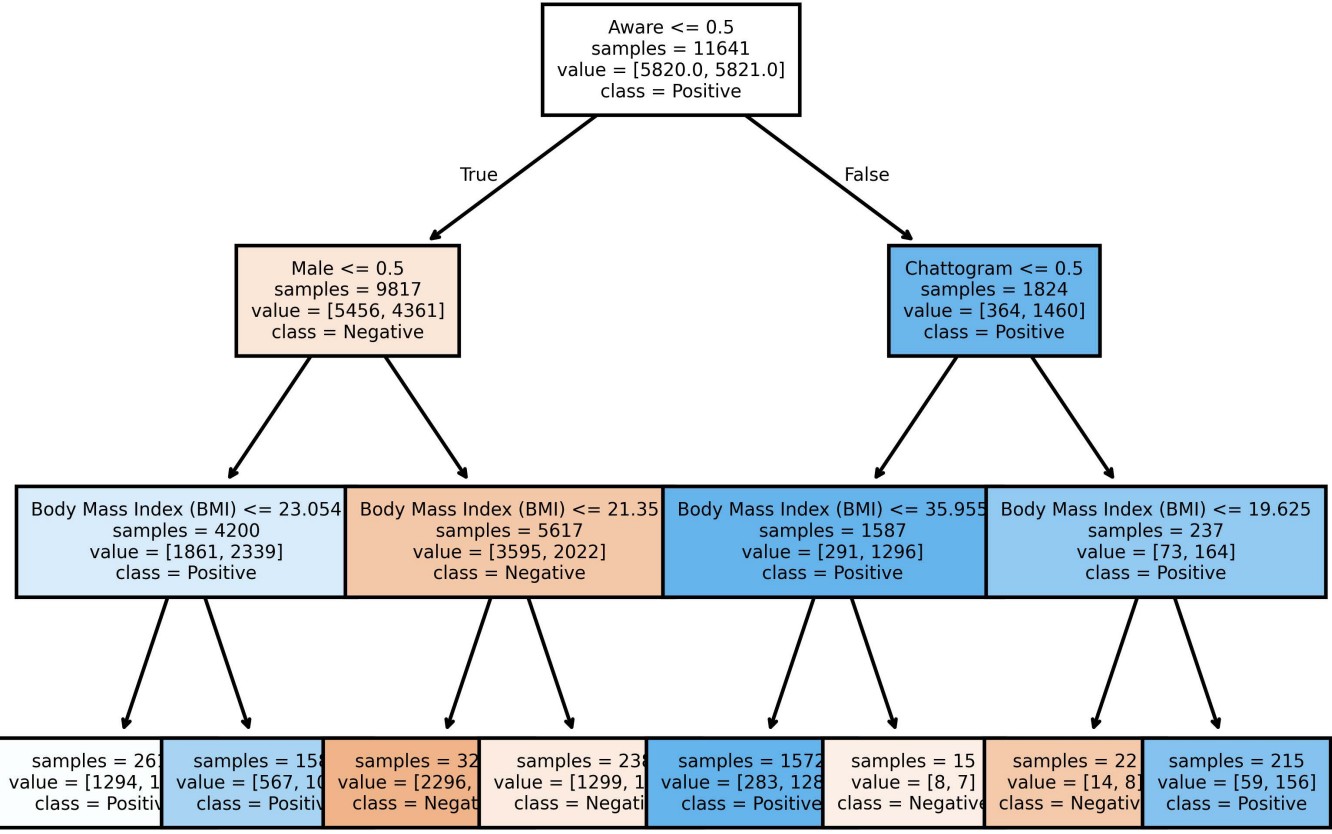

**Fig 5. Decision Tree shows this process is classifying participants into two different classes based on their several characteristics.** It's structure where an internal node represents a feature, the branch represents a decision rule, and each leaf node represents the outcome.

index (BMI) below 22 is predicted to be non-hypertensive. This straightforward decision-making process enhances model interpretability.

## Interaction factors

The Individual Conditional Expectation (ICE) plot of hypertension probability by age, shown in Fig 6, indicates that for some individuals with a predicted hypertension probability above 0.5, the prediction remains relatively unchanged at higher ages. Compared to age 35, predictions for most individuals show little change until age 50, at which point the predicted probability begins to increase. Fig 7 presents an ICE plot between partial dependence predictive values and body mass index (BMI), showing that for some individuals with a predicted hypertension probability above 0.5, the predictions for most participants remain unchanged until a BMI of 24, after which the predicted probability increases. In contrast, the ICE plot of BMI probability by age, depicted in Fig 8, reveals that the predicted high BMI probability for most participants decreases as age increases. Additionally, it shows that the predictive values of BMI higher in female and self-awareness. Similar to the age effect on hypertension, predictions remain stable until age 50, beyond which the probability begins to decline. Furthermore, the partial predicted probability of hypertension by sex, displayed as a boxplot in Fig 9, demonstrates that the median predicted probability of hypertension was higher among females compared to those males.

SHAP values visualization in Fig 10 highlight the contribution of individual factors to hypertension predictions. The body mass index (BMI) is identified as the most influential predictor, followed by self-awareness, sex, and age, while household

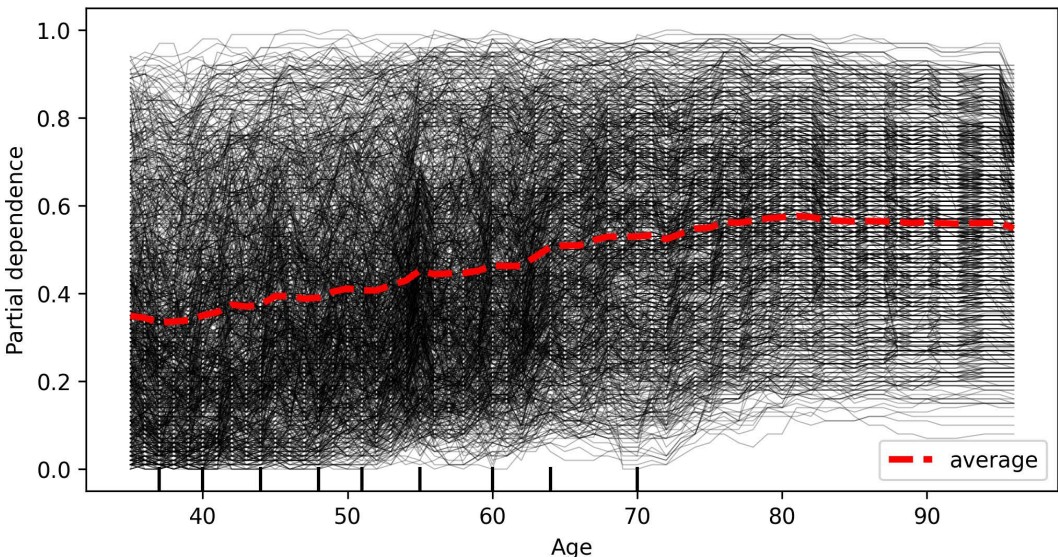

**Fig 6. Individual Conditional Expectation (ICE) plot presenting the predicted probability of hypertension across age.** Each line represents an individual participant. The plot illustrates how predicted hypertension risk changes with age while holding other variables constant. For most participants, the predicted probability of hypertension increases with increasing age, indicating a consistent positive relationship between age and hypertension risk in the model.

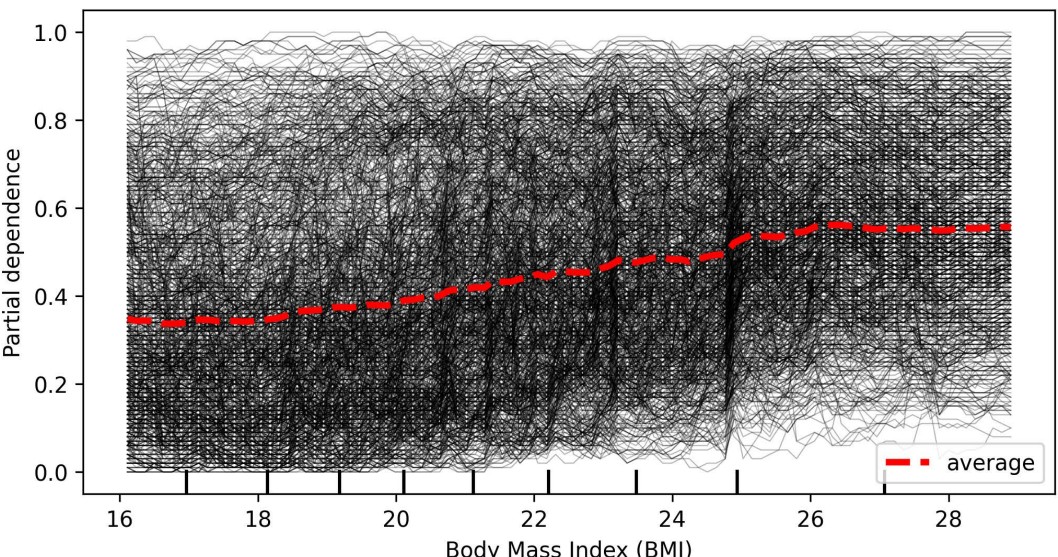

**Fig 7. Individual Conditional Expectation (ICE) plot showing the predicted probability of hypertension across BMI.** Each line represents an individual participant. The plot illustrates how predicted hypertension risk changes with body mass index while holding other variables constant. For most participants, the predicted probability of hypertension increases with increasing body mass index (BMI), indicating a consistent positive relationship between BMI and hypertension risk in the model.

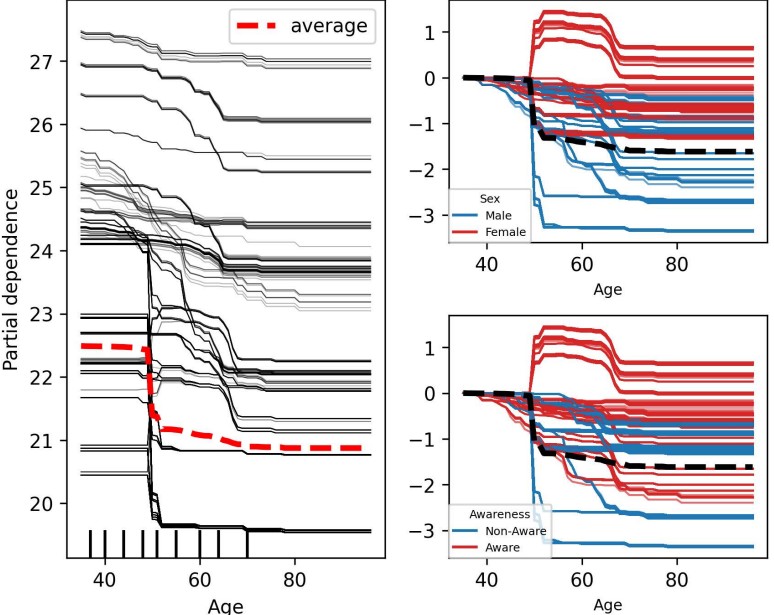

**Fig 8. Individual Conditional Expectation (ICE) plot showing the predicted probability of BMI across age, sex and self-awareness.** Each line represents an individual participant. The plot illustrates how predicted BMI changes with age, sex and self-awareness while holding other variables constant. For most participants, the predicted probability of BMI decreases with increasing age, male and non-aware.

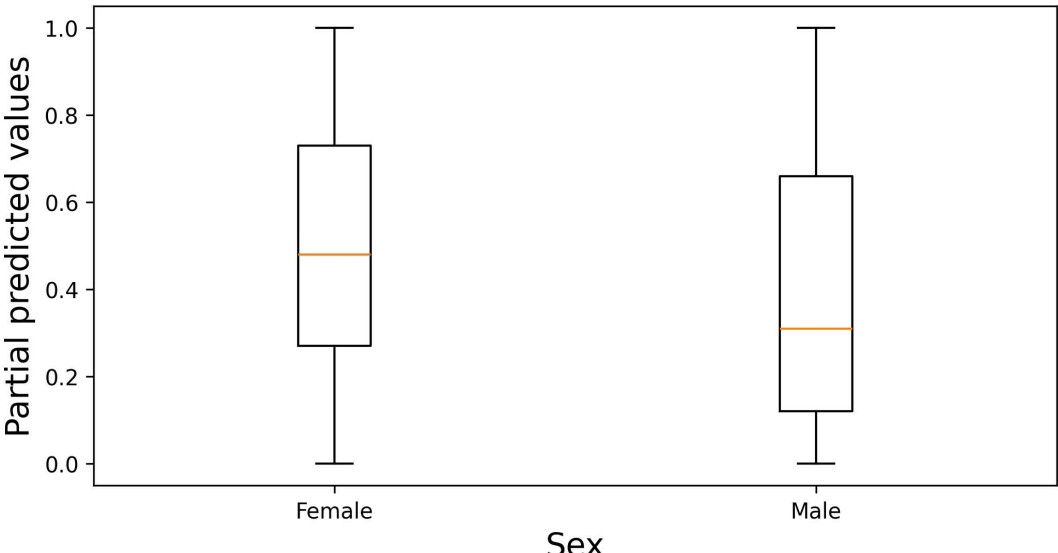

**Fig 9. Dependency boxplot showing the relationship between sex and the partial predicted probability of hypertension.** Boxes represent the interquartile range (IQR), with the median indicated by the central line; whiskers extend to 1.5 times the IQR, and outliers are plotted as individual points.

wealth and educational attainment rank as the least influential. A scatter plot of SHAP values for hypertension predictions in Fig 11 demonstrates that excessive body weight, self-awareness, being female, and older age significantly contribute to hypertension predictions. Additionally, Fig 12 reveals that age, body mass index (BMI), sex, residential areas, and self-awareness act as interaction factors and exhibit nonlinear, complex relationships in predicting hypertension.

PLOS Computational Biology logo

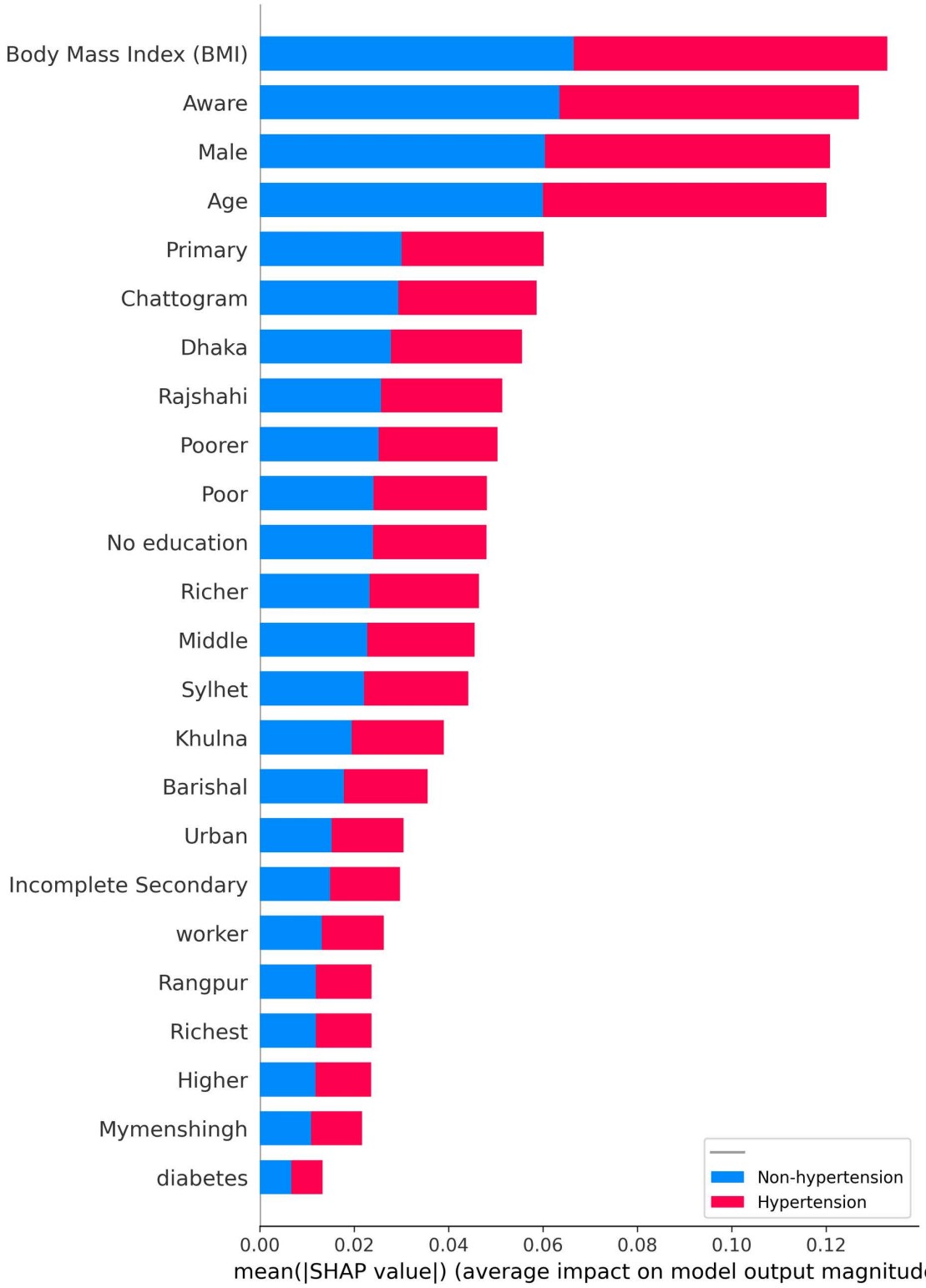

**Fig 10. Feature importance based on SHAP values, depicted as a bar plot.** Each bar represents a feature (Y-axis), and its length along the X-axis indicates the mean absolute SHAP value, signifying the average magnitude of impact on the model's log-odds output. The color encoding reflects the feature's value for a given prediction: red for values pushing towards a hypertension prediction and blue for values pushing towards a non-hypertension prediction.

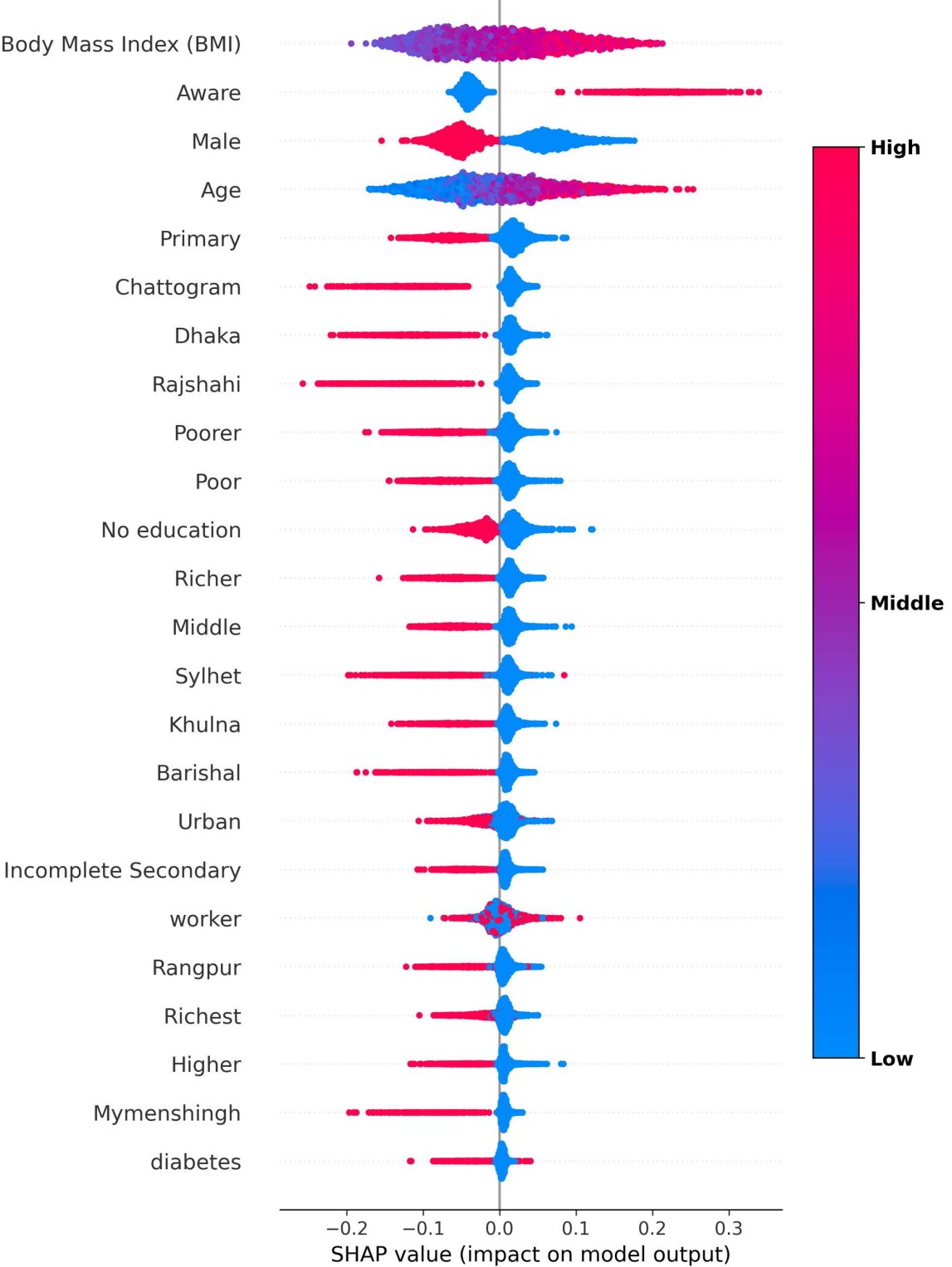

**Fig 11. SHAP scatter plot illustrating feature contributions to hypertension predictions.** It highlights features associated with an increased likelihood of hypertension (typically positive SHAP values) and non-hypertension (typically negative SHAP values), while also identifying features with minimal impact (SHAP values near zero).

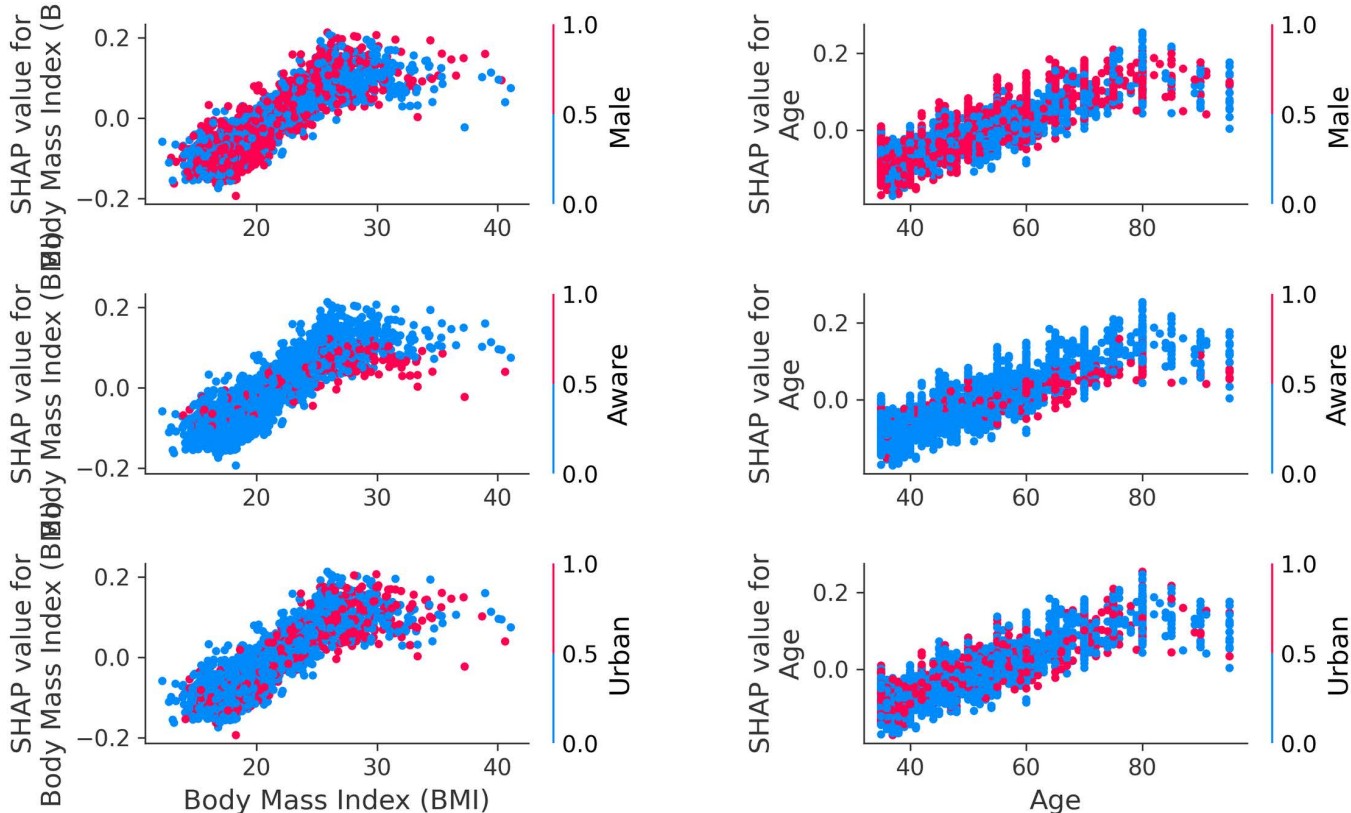

**Fig 12. Dependency plot of SHAP values for hypertension, showing the relationship between a feature's value (X-axis) and its corresponding SHAP value (Y-axis, indicating impact on prediction).** Point coloration by a second feature visualizes how interactions between these two features affect the predicted outcome.

## Estimating causal association with hypertension

Based on the predefined causal structure shown in Fig 1, Table 4 summarizes the crude odds ratio (OR) of excessive body weight is 2.24 (95%CI: 2.07 – 2.42). After adjusted for age, sex, SES, and self-awareness, the adjusted OR is 1.97 (95%CI: 1.79 – 2.17). The reduction in the OR from 2.24 to 1.97 suggests that some of the initial association was indeed attributable to these confounding factors. By using Double Machine Learning, the OR is 1.30 (95%CI: 1.17 – 1.43), revealed a robust causal effect of excessive body weight on hypertension.

**Table 4. Observational causal association between excessive body weight on hypertension, including crude, adjusted for age, sex, SES, self-awareness by using logistic regression, and Double Machine Learning with Random forest model among participants aged 34 year and older in Bangladesh from 2011 to 2022.**

| Models | Odds Ratio (OR) | 95%CI | p-values |
|---|---|---|---|
| Univariate logistic regression - Crude | 2.24 | 2.07 – 2.42 | <0.01 |
| Multivariate logistic regression – Adjusted | 1.97 | 1.79 – 2.17 | <0.01 |
| Double Machine Learning (DML) | 1.30 | 1.17 – 1.43 | <0.01 |

## Discussion

This study used a machine learning models with high predictive accuracy to identify the most important predictors of hypertension based on individual's characteristics, household characteristics, socioeconomic status, self-awareness, diabetes, and nutritional status. The Random forest classifier demonstrated the best performance among the evaluated models, achieving the lowest training and validation errors. This supervised machine learning model effectively captured patterns in the data while ensuring accurate predictions, making it the most suitable model for the given data. The most influential predictors of hypertension included body mass index (BMI), age, socioeconomic status, self-awareness of hypertension, sex, residence, working status, and diabetes, while educational status had a lesser effect on prediction of hypertension. The analysis revealed complex, nonlinear interactions among predictors such as body mass index (BMI), age, sex, self-awareness, and residential areas with hypertension outcomes. By using Double Machine Learning models based on pre-defined causal structure, the study also identified the causal effect of several predictors. Specifically, the causal effect of excess body weight on adult hypertension risk was found to be confounded by other predictors [18,38,52–54].

Machine learning -based real-time predictive systems offer promising applications for hypertension management [55]. These tools can support preliminary diagnosis, reduce hospital readmission, and lower healthcare costs while mitigating the progression of severe illness. Beyond serving as decision support tools for physicians, predictive diagnostic systems can be employed in hospital triage units to assist nurse practitioners or by patients at home. For example, wearable technologies and Bluetooth-enabled devices like blood pressure monitors can transmit vital data to predictive models, providing real-time health assessments. Such innovations could empower patients to seek timely medical interventions, preventing critical situations and promoting better health outcomes [53,55].

It is noteworthy that adult hypertension was predicted by key factors such as excessive body weight, older age, self-awareness of hypertension, being female, and urban residency—factors that remain directly or indirectly associated with hypertension despite the implementation of the HPNSDP. For example, unhealthy lifestyle behaviors like smoking, increased levels of education, urbanization, and greater self-awareness of hypertension status have contributed to a reduction in under-weight prevalence but have simultaneously heightened exposure to excessive body weight. Consequently, while the risk associated with poor quality of lifestyles, urban living, and socioeconomic status have decreased, the contribution of excessive body weight to hypertension risk has grown. Excess calorie intake is stored as fat tissue, leading to weight gain, and these fat deposits contribute to elevated cholesterol levels in the blood, ultimately increasing blood pressure [56].

Since 2011, increased self-awareness of non-communicable diseases (NCDs) has contributed to a rise in hypertension prevalence, partly due to more frequent monitoring of blood pressure. Wealthier families, with their ability to adopt healthier diets and behaviors, have shown a greater predictive probability of hypertension and its association with nutritional status. Consequently, excessive body weight has emerged as a key predictor with a strong positive association with hypertension [25,57,58]. Previous studies have demonstrated a strong link between lower socioeconomic status (SES) and hypertension in high-income countries, largely influenced by self-awareness and better hypertension management. However, this relationship is more complex and sometimes inconsistent in low-income countries. A notable finding is that self-awareness was a significant predictor of hypertension, whereas SES showed a weaker association. Excessive body weight was identified as the most influential predictor of hypertension, followed by older age and self-awareness [18,38,59]. In line with earlier research from the same country, these factors were confirmed as significant predictors for hypertension. Additionally, we found that predictors such as sex and residential area interacted with excessive body weight and hypertension, acting as confounders in predicting hypertension. These findings emphasize the relevance of awareness programs aimed at reducing excessive body weight to mitigate hypertension risk [15,38,39,60].

The machine learning (ML) study on hypertension by Islam et al.[53] focused on predictive modeling, employing algorithms such as XGBoost, Random Forest, and Logistic Regression to identify individuals at risk and highlight important predictors based on associations. These models were assessed using standard evaluation metrics such as accuracy,

precision, and recall, with feature importance tools used to identify influential factors like age and BMI. However, these models reveal correlations rather than causal relationships, and the rankings reflect associations rather than cause-and-effect, particularly in the context of South Asian populations. Notably, excessive body weight emerged as a strong predictor of hypertension using traditional ML methods, but these results cannot establish causality. In contrast, the current study applied the Double Machine Learning (DML) approach using the Bangladesh Demographic and Health Survey (BDHS) dataset, specifically designed for causal inference. By incorporating a Directed Acyclic Graph (DAG), the DML framework identified excessive body weight not only as a predictor but as a causal factor for hypertension, separating its effect from confounding influences through cross-fitting and orthogonalization. While traditional ML models are well-suited for risk prediction and classification, DML offers a more rigorous understanding of causal mechanisms, making it a better choice when the goal is to assess how strongly excessive body weight contributes to hypertension as an outcome [53].

SHAP value visualizations identified socioeconomic status (SES), age, sex, self-awareness, and nutritional status as important predictors associated with hypertension. Additionally, Using a causal diagram [18], we identified socioeconomic status (SES), age, sex, and self-awareness of hypertension as confounders in the relationship between nutritional status and hypertension. This indicates that nutritional status does not have a direct causal effect on hypertension but is instead correlated with these confounders, which drive the likelihood of hypertension. Our predictive model highlights excessive body weight as the strongest single predictor of hypertension, as it encapsulates the influence of many true causal drivers through its correlations. Random forest model employs regularization to select the simplest possible model that maintains high predictive accuracy, minimizing overfitting. If nutritional status alone could predict hypertension as effectively as SES, age, sex, and self-awareness, the model would prefer it. However, when nutritional status is highly correlated with these confounders, the machine learning model tends to favor independent predictors over directly causal factors to enhance robustness in its predictions [19,52]. To improve hypertension prediction, it is essential to identify and manipulate the factors that genuinely influence outcomes. Using Double Machine Learning techniques, we estimated the unbiased causal effect of excessive body weight on hypertension while controlling for age, sex, self-awareness, and SES [61]. This analysis revealed a significant causal effect of excessive body weight on adult hypertension, underscoring that it may not act as an entirely independent predictor. Therefore, interventions targeting excessive body weight are critical for reducing hypertension risk [18,62,63].

This study has several strengths. First, the machine learning prediction modeling utilized a population-based, nationally representative survey dataset, making the findings broadly applicable to various populations and settings. Additionally, field workers employed well-established digital techniques to measure blood pressure and anthropometric parameters, ensuring high-quality data collection. The use of Double Machine Learning models enhanced the precision of hypertension predictions by leveraging observational causal inference from cross-sectional survey data. Second, given the dataset size of 11,815 samples, it is important to reflect on the computational complexity and practical feasibility of the methods employed in this study. The Random Forest algorithm, while efficient and parallelizable, has a time complexity of approximately $O(ntree \times n \times log\ n)$, where $n$ is the number of samples and $ntree$ is the number of trees. In our case, with 500 trees and a moderate sample size, model training remained computationally manageable. SHAP (SHapley Additive Explanations), known for its interpretability, is more computationally intensive due to its reliance on game-theoretic principles; however, the use of TreeSHAP optimized for tree-based models substantially reduced the burden, making it feasible for our dataset. ICE (Individual Conditional Expectation) plots were generated for selected variables only, which limited their runtime. The Double Machine Learning (DML) approach, while more computationally demanding due to its cross-fitting and orthogonalization steps, was effectively implemented using efficient Python libraries. Overall, all methods were executed on standard computing infrastructure without the need for high-performance computing, indicating good scalability and applicability to similar population-based datasets. This demonstrates the practical potential of these methods for researchers and public health practitioners working with medium-sized datasets in resource-constrained settings.

However, the study also has limitations. The cross-sectional design limits the ability to infer cause-and-effect relationships. Hypertension is measured using a single blood pressure reading and self-reported awareness of hypertension status, which can introduce bias into the assessment. Moreover, crucial data on factors strongly linked to hypertension, such as diet, salt intake, exercise levels, saturated fat consumption, sleep patterns, and drinking alcohol habit were not collected, preventing their inclusion in the machine learning models. These unmeasured confounders such as salt intake, alcohol consumption, sleep quality, stress, and genetic predispositions, as well as the reliance on a single blood pressure measurement. these are significant factors in the etiology of hypertension. Consequently, confounding by these unmeasured factors could not be addressed in the model.

Our prediction analysis was restricted to observed confounders in three distinct population samples, but sampling errors and biases from unobserved confounders could still affect the results. While Double Machine Learning model offers the advantage of learning complex patterns and may be more robust to unmeasured confounding than traditional logistic regression, residual confounding remains a potential source of bias in the findings.

## Conclusion

This study highlights the value of machine learning models, particularly Random forest model and Double Machine Learning techniques, in identifying key predictors of hypertension using nationally representative data. Excessive body weight emerged as the most influential predictor, followed by age, self-awareness of hypertension, and socioeconomic status. While these associations are well-established in the literature, their complex interdependencies highlight the importance of appropriately addressing confounding in observational analyses. The main contribution of this work lies in the methodological application of DML, which enables more rigorous causal inference in the presence of nonlinear relationships and high-dimensional confounders. By leveraging this approach, we estimated the potential causal effect of excessive body weight on hypertension with greater robustness than conventional methods. Rather than focusing solely on known associations, our study illustrates how DML can serve as a powerful tool for causal analysis in public health research where randomized trials are not feasible. The Double Machine Learning approach proved especially effective in identifying the potential causal effect of excessive body weight on hypertension. Its generalizability makes it applicable to a wide range of public health challenges, offering valuable insights for evidence-based decision-making and the design of targeted interventions. The study findings demonstrate the potential of machine learning tools in enabling hypertension prediction in future and guiding targeted prevention strategies. To enhance the applicability and predictive power of such models, future research should incorporate more comprehensive datasets, include additional variables such as dietary salt intake, alcohol consumption, sleep quality, psychological stress, and genetic predisposition, and adopt longitudinal study designs. These advancements will further refine our ability to model and prevent hypertension and improve outcomes through better resource allocation for at-risk populations. This study recommends continued methodological innovation and thoughtful integration into policy and practice.

## Acknowledgments

The authors would like to acknowledge the program of Demographic and Health Survey (DHS) team for providing survey data. The author(s) would like to acknowledge the contribution of caregivers/ mothers of the study participants for their consent to enroll children in the study. icddr,b is also grateful to the Governments of Bangladesh and Canada, for providing core/unrestricted support.

## Author contributions

**Conceptualization:** Probir Kumar Ghosh, Md. Aminul Islam, Md. Ahshanul Haque, Md. Tariqujjaman, Novel Chandra Das, Mohammad Ali, Md. Rasel Uddin, Md. Golam Dostogir Harun.

**Data curation:** Probir Kumar Ghosh, Md. Aminul Islam, Md. Ahshanul Haque, Md. Tariqujjaman, Novel Chandra Das, Mohammad Ali, Md. Rasel Uddin, Md. Golam Dostogir Harun.

**Formal analysis:** Probir Kumar Ghosh, Md. Aminul Islam, Md. Ahshanul Haque, Md. Tariqujjaman, Novel Chandra Das, Mohammad Ali, Md. Rasel Uddin, Md. Golam Dostogir Harun.

**Investigation:** Probir Kumar Ghosh, Md. Aminul Islam, Md. Tariqujjaman, Novel Chandra Das, Mohammad Ali, Md. Rasel Uddin, Md. Golam Dostogir Harun.

**Methodology:** Probir Kumar Ghosh, Md. Aminul Islam, Md. Ahshanul Haque, Md. Tariqujjaman, Novel Chandra Das, Mohammad Ali, Md. Rasel Uddin, Md. Golam Dostogir Harun.

**Project administration:** Probir Kumar Ghosh.

**Resources:** Probir Kumar Ghosh.

**Software:** Probir Kumar Ghosh, Md. Aminul Islam, Md. Ahshanul Haque, Md. Tariqujjaman, Novel Chandra Das, Mohammad Ali, Md. Rasel Uddin, Md. Golam Dostogir Harun.

**Supervision:** Probir Kumar Ghosh, Md. Golam Dostogir Harun.

**Validation:** Probir Kumar Ghosh, Md. Aminul Islam, Md. Ahshanul Haque, Md. Tariqujjaman, Novel Chandra Das, Mohammad Ali, Md. Rasel Uddin, Md. Golam Dostogir Harun.

**Visualization:** Probir Kumar Ghosh, Md. Aminul Islam, Md. Ahshanul Haque, Md. Tariqujjaman, Novel Chandra Das, Mohammad Ali, Md. Rasel Uddin, Md. Golam Dostogir Harun.

**Writing – original draft:** Probir Kumar Ghosh, Md. Aminul Islam, Md. Ahshanul Haque, Md. Tariqujjaman, Novel Chandra Das, Mohammad Ali, Md. Rasel Uddin, Md. Golam Dostogir Harun.

**Writing – review & editing:** Probir Kumar Ghosh, Md. Aminul Islam, Md. Ahshanul Haque, Md. Tariqujjaman, Novel Chandra Das, Mohammad Ali, Md. Rasel Uddin, Md. Golam Dostogir Harun.

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
