## [Decision Letter · Decision Letter 0]

Identifying predictors and assessing causal effect on hypertension risk among adults using double machine learning models: insights from Bangladesh Demographic and Health Survey

PLOS Computational Biology

Dear Dr. Ghosh,

Thank you for submitting your manuscript to PLOS Computational Biology. After careful consideration, we feel that it has merit but does not fully meet PLOS Computational Biology's publication criteria as it currently stands. Therefore, we invite you to submit a revised version of the manuscript that addresses the points raised during the review process.

Please submit your revised manuscript within 60 days Jul 08 2025 11:59PM. If you will need more time than this to complete your revisions, please reply to this message or contact the journal office at ploscompbiol@plos.org. Please include the following items when submitting your revised manuscript:

We look forward to receiving your revised manuscript.

Kind regards,

Pingzhao Hu

Academic Editor

PLOS Computational Biology

Jennifer Flegg

Section Editor

PLOS Computational Biology

**Journal Requirements:**

1) Please provide an Author Summary. This should appear in your manuscript between the Abstract (if applicable) and the Introduction, and should be 150-200 words long. The aim should be to make your findings accessible to a wide audience that includes both scientists and non-scientists. Sample summaries can be found on our website under Submission Guidelines:

2) Please confirm whether the tables included in the manuscript are main ones or supplementary. If they are main tables, they should be included in the manuscript. If the tables are supplementary, they should not be incorporated in the manuscript ;however, they should be uploaded as separate file with the file type 'Supporting Information'.

Note : Supplementary tables should be cited and labeled as "S1 Table" , "S2 Table" and so forth. Please ensure that each Supporting Information file has a legend listed in the manuscript after the references list.

3) Please note that your Data Availability Statement is currently missing the repository name. If your manuscript is accepted for publication, you will be asked to provide these details on a very short timeline. We therefore suggest that you provide this information now, though we will not hold up the peer review process if you are unable.

**Reviewers' comments:**

Reviewer's Responses to Questions

**Comments to the Authors:**

**Please note that two reviews are uploaded as attachments.**

Reviewer #1: The manuscript, "Identifying Predictors and Assessing Causal Effects on Hypertension Risk Among Adults Using Double Machine Learning Models: Insights from the Bangladesh Demographic and Health Survey," thoroughly examines the factors influencing hypertension and their causal effects. Notably, the authors demonstrate that BMI and self-awareness significantly impact hypertension, with a higher BMI increasing the likelihood of hypertension. However, the results would benefit from further validation and clarification of key points.

Major

1. The study primarily employs the double/de-biased machine learning model, yet its methodological details are insufficiently explained. Given that causal inference from this approach is central to the study’s conclusions, a more detailed explanation of the double/de-biased machine learning process would strengthen the study’s reliability. For example, please include (1) the basic methodological concept of double/de-biased machine learning model introduced in ref (36) and (2) the performance of that model in previous studies.

2. The model performance in predicting hypertension risk was assessed by randomly splitting the dataset into training and test sets. This randomness introduces the possibility that different random seeds could yield varying prediction errors. To ensure robustness, we recommend the following validation procedure

- Perform 10,000 iterations (or a sufficiently large number) of train-test splits.

- Compute performance metrics (e.g., accuracy, error, precision) for each split.

- Report the distribution of these metrics or representative statistics such as mean, median, and standard deviation.

Minor

1. Figure 2: In line 208, the training and validation errors for the random forest model are reported as 0.03% and 0.65%, respectively, suggesting an accuracy (i.e. 100% – error (%)) exceeding 99%. However, the confusion matrix in Figure 2 indicates an accuracy of 93%. Could the authors clarify the reason for this discrepancy?

2. Figure 3: Feature importance can be quantified using different measures, such as frequency (or weight), gain, or total gain. Which measure was used in Figure 3?

Reviewer #2: The study titled: ‘Identifying predictors and assessing causal effect on hypertension risk among adults using double machine learning models: insights from Bangladesh Demographic and Health Survey’ utilizes machine learning (ML) and causal inference methods (specifically double/de-biased machine learning) to identify predictors and estimate the causal effect of excessive body weight on hypertension among Bangladeshi adults aged 34 and above, using nationally representative data from BDHS 2011 and 2022.

The study offers some novelty and notable strengths, employs a fairly large dataset (n = 11,815), and addresses a subject that is both valuable for exploration and well-justified from a public health perspective.

It applies advanced ML methods with the benefit of random forest model over the rest.

However, there are few major issues that should be addresses.

1. Overfitting risk not adequately addressed

Claim: Random Forest achieves 93% accuracy with only 0.03% training error and 0.65% validation error.

Issue: This discrepancy may suggest possible overfitting, especially when such low training error is achieved in a high-dimensional, non-simulated real-world dataset.

While the use of SMOTE is appropriate for addressing class imbalance, it is not sufficient on its own to prevent overfitting (statement in lines 163-164). Without complementary measures such as proper cross-validation, regularization, or ensuring that synthetic samples are excluded from the test set, the reported model performance may be overly optimistic and not fully reliable. Especially when trained to near-perfect performance (as in this study: 0.03% training error), the model may memorize noise or artifacts — even with balanced classes. Moreover, SMOTE was applied to both training and test sets. This violates standard evaluation protocols, as synthetic instances should never be in the test set. This inflates performance and masks overfitting.

Possible fix: Apply cross-validation and report results across folds. Include ROC-AUC, precision-recall curves, and/or calibration metrics.

2. The manuscript refers to “double/de-biased random forest”, but that’s a misunderstanding:

No clear mention of how nuisance functions were estimated or cross-fitting implementation (key DML requirements) and how exactly this method was implemented, please elaborate in refference to [36].

3. Lines 180-183.

The causal graph (DAG) (Figure 12) is briefly mentioned but lacks detail and validation:

No justification of causal assumptions (e.g., exclusion restrictions, conditional independencies).

No robustness checks (e.g., sensitivity analysis for unobserved confounding).

Possible fix: Expand on causal identification strategy, include discussion of DAG assumptions, and/or use tools like DAGitty or do-calculus if appropriate.

4. The role of self-awareness is conceptually ambiguous throughout the manuscript. It is simultaneously treated as a confounder, a mediator, and an interaction factor, without a clear causal justification or supporting DAG structure. Clarifying whether self-awareness is a source of bias (confounder), part of the causal mechanism (mediator), or modifies the effect of other predictors (interaction) is essential for accurate interpretation of results and proper model specification. Please clarify. It seems, like SES is treated as a confounder and predictor, but the mediation paths are not analyzed.

Possible fix: Clarify variable roles—draw a more detailed DAG (Fig. 12) and justify each path.

5. The authors acknowledge limitations but fail to sufficiently address implications of:

no data on salt intake, alcohol, sleep, stress, genetics, etc., single blood pressure measurement (instead of multiple averaged readings). These are major unmeasured confounders in hypertension etiology, which undermines the claim of causal inference.

Possible fix: Consider proxy variables or conduct sensitivity analysis (e.g., Rosenbaum bounds or E-values), and explicitly mention this caveat in the abstract. Minimaly it merits to address those in Discussion.

6. SHAP value interpretations are sometimes inconsistent:

SHAP shows association, not causality. The manuscript often blurs this distinction. The narrative implies BMI causes hypertension because SHAP ranks it highest—this is misleading without strict causal validation.

Possible fix: Separate predictive vs causal interpretations clearly.

7. Baseline Comparison: No baseline prevalence-adjusted model reported—how does RF improve over age/BMI alone? Ref. Fig. 3.

8. The role of self-awareness is conceptually ambiguous throughout the manuscript. It is simultaneously treated as a confounder, a mediator, and an interaction factor, without a clear causal justification or supporting DAG structure. Clarifying whether self-awareness is a source of bias (confounder), part of the causal mechanism (mediator), or modifies the effect of other predictors (interaction) is essential for accurate interpretation of results and proper model specification. Please clarify. It seems, like SES is treated as a confounder and predictor, but the mediation paths are not analyzed.

Possible fix: Clarify variable roles—draw a more detailed DAG and justify each path.

9. Some minor issues:

For SHAP, authors used incorrect expansion of the abbreviation. It should be Shapley, instead of Shapely (line 30, Abstract).

Language correction is required, there are some typos, e.g. line 156 ‘mode’ instead of ‘model’, awkward phrasing, inconsistent tense (e.g., “was analyzed them”).

Redundancy: The study repeatedly states similar findings (e.g., BMI as strongest predictor) without adding depth.

Conclusion overclaims: The conclusion suggests policy implications based on causal findings, which is risky without robust causal validation.

Reviewer #3: A detailed review has been uploaded as an attachment. In summary, the manuscript addresses an important topic but requires significant improvements in clarity, methodological detail, and justification of novelty. Specific major and minor comments are provided in the attached review to assist the authors in strengthening the work.

**Have the authors made all data and (if applicable) computational code underlying the findings in their manuscript fully available?**

Reviewer #1: **No: ** The code underlying this study (e.g. training ML, SMOTE algorithm, ...) is not explicitly available in the manuscript

Reviewer #2: Yes

Reviewer #3: **No: ** The manuscript does not provide sufficient information regarding the availability of the data and computational code used in the study. It is not clear whether the processed data, modeling code (e.g., for DML, Random Forest, SHAP/ICE analyses), or preprocessing steps have been made publicly accessible. To comply with PLOS's data availability policy and enhance reproducibility, I recommend that the authors clearly state where the data and code can be accessed or, if there are restrictions (e.g., third-party ownership or participant confidentiality), explain them explicitly in the Data Availability Statement. Sharing data and code through a public repository such as GitHub, Dryad, or Zenodo would be ideal.

PLOS authors have the option to publish the peer review history of their article (what does this mean? ). If published, this will include your full peer review and any attached files.

**Do you want your identity to be public for this peer review?** For information about this choice, including consent withdrawal, please see our Privacy Policy .

Reviewer #1: No

Reviewer #2: No

Reviewer #3: No

**Figure resubmission:**

**Reproducibility:**



---

## [Decision Letter · Decision Letter 1]

PCOMPBIOL-D-25-00212R1

Identifying predictors and assessing causal effect on hypertension risk among adults using Double machine learning models: insights from Bangladesh Demographic and Health Survey

PLOS Computational Biology

Dear Dr. Ghosh,

Thank you for submitting your manuscript to PLOS Computational Biology. After careful consideration, we feel that it has merit but does not fully meet PLOS Computational Biology's publication criteria as it currently stands. Therefore, we invite you to submit a revised version of the manuscript that addresses the points raised during the review process.

Please submit your revised manuscript within 30 days Aug 04 2025 11:59PM. If you will need more time than this to complete your revisions, please reply to this message or contact the journal office at ploscompbiol@plos.org. Please include the following items when submitting your revised manuscript:

We look forward to receiving your revised manuscript.

Kind regards,

Pingzhao Hu

Academic Editor

PLOS Computational Biology

Jennifer Flegg

Section Editor

PLOS Computational Biology

**Reviewers' comments:**

Reviewer's Responses to Questions

**Comments to the Authors:**

Reviewer #1: The authors completely addressed all comments

Reviewer #2: The authors addressed all the concerns. Thank you very much.

Reviewer #3: Major revision

In the conclusion section (lines 466–467), the manuscript claims that the Double Machine Learning approach offers actionable insights for policy development regarding the impact of excessive body weight on hypertension. However, the manuscript does not clearly articulate what those actionable insights are. Moreover, the association between excessive body weight and hypertension is already well-established in the literature, as acknowledged through the citations included.

The true contribution of this work lies more in the methodological approach—specifically, the use of Double Machine Learning to estimate causal effects in complex, observational settings. This novelty and its potential for generalization to other public health areas should be emphasized. The authors should revise the conclusion to better reflect this, by:

Highlighting how DML strengthens causal inference in the presence of nonlinear confounding.

Discussing how the approach can be generalized and applied to other datasets or conditions.

Offering clearer guidance on how methodological advancements could inform future research or broader health interventions, beyond the known association with body weight.

Minor Revision

There is a typographical issue in line 172 — an unmatched or stray closing bracket “])”. Please correct this to ensure consistency in formatting and readability.

The term Double Machine Learning appears with inconsistent capitalization throughout the manuscript (e.g., "Double Machine Learning" vs. "Double machine learning"). Please standardize the format and use Double Machine Learning consistently across the entire document, including the title, abstract, methods, and discussion sections.

**Have the authors made all data and (if applicable) computational code underlying the findings in their manuscript fully available?**

Reviewer #1: None

Reviewer #2: Yes

Reviewer #3: None

PLOS authors have the option to publish the peer review history of their article (what does this mean? ). If published, this will include your full peer review and any attached files.

**Do you want your identity to be public for this peer review?** For information about this choice, including consent withdrawal, please see our Privacy Policy .

Reviewer #1: No

Reviewer #2: No

Reviewer #3: No

**Figure resubmission:**
---

## [Editor Report · Decision Letter 2]

Dear Ghosh,

We are pleased to inform you that your manuscript 'Identifying predictors and assessing causal effect on hypertension risk among adults using Double Machine Learning models: insights from Bangladesh Demographic and Health Survey' has been provisionally accepted for publication in PLOS Computational Biology.

Best regards,

Pingzhao Hu

Academic Editor

PLOS Computational Biology

Jennifer Flegg

Section Editor

PLOS Computational Biology

---

## [Editor Report · Acceptance letter]

PCOMPBIOL-D-25-00212R2

Identifying predictors and assessing causal effect on hypertension risk among adults using Double Machine Learning models: insights from Bangladesh Demographic and Health Survey

Dear Dr Ghosh,

I am pleased to inform you that your manuscript has been formally accepted for publication in PLOS Computational Biology. Your manuscript is now with our production department and you will be notified of the publication date in due course.

With kind regards,

Zsuzsanna Gémesi
